# Performance and Meat Quality of Dual-Purpose Cockerels of Dominant Genotype Reared on Pasture

**DOI:** 10.3390/ani10030387

**Published:** 2020-02-27

**Authors:** Michaela Englmaierová, Miloš Skřivan, Tomáš Taubner, Věra Skřivanová

**Affiliations:** Department of Nutrition Physiology and Animal Product Quality, Institute of Animal Science, 104 00 Prague-Uhrineves, Czech Republic; skrivan.milos@vuzv.cz (M.S.); taubner.tomas@vuzv.cz (T.T.); skrivanova.vera@vuzv.cz (V.S.)

**Keywords:** layer line chickens, sensory analysis, fatty acids, oxidative stability, α-tocopherol

## Abstract

**Simple Summary:**

One-day-old laying cockerels are killed after hatching because they do not reach the growth rate of broiler chickens, and their fattening would be economically disadvantageous. A possible variant of the use of these cockerels could be organic or free-range farming, where it is desirable that the animals are fattened for a longer period of time, are more physically active, and graze pasture vegetation. Another possibility is dual-purpose genotype breeding, where hens are used for egg production and cockerels are fattened for meat. In the present study, three genotypes of dual-purpose cockerels Dominant were compared. All three genotypes showed the ability to graze with resulting improvements in meat quality. The Dominant 102 cockerels have the greatest prerequisites for use in extensive fattening, mainly due to higher performance, willingness to graze and vitamin E content, which protects the fat from becoming rancid.

**Abstract:**

The culling of layer cockerels due to economic inefficiency is an ethical problem. Organic or free-range fattening of these cockerels or dual-purpose genotypes breeding is a possible solution to this problem. The aim of the study was to assess the differences in performance and meat quality characteristics in dual-purpose cockerels Dominant of three genotypes (Dominant Sussex D 104, Dominant Brown D 102 and Dominant Tinted D 723, 100 cockerels per genotype) with access to pasture. The cockerels were housed in mobile boxes on the pasture herbage from the 50th to 77th day of age (stocking density: 0.108 m^2^/bird). The highest body weight on the 77th day of age (*p* < 0.001) and the nonsignificantly lowest feed conversion was achieved by Dominant Brown D 102 cockerels (1842 g and 2.79, respectively). Non-significantly higher pasture herbage intake on the 70th day of age was recorded in genotype Dominant Brown D 102 (7.41 g dry matter (DM)/bird/day) and Dominant Tinted D 723 (7.52 g DM/bird/day). The pasture herbage contained 56.9 mg/kg DM α-tocopherol, 170.3 mg/kg DM zeaxanthin and 175.0 mg/kg DM lutein and had a favourable n6/n3 ratio (0.26). The boiled meat of cockerels Dominant Tinted D723 showed the highest tenderness based on both the sensory evaluation (*p* = 0.022) and the value of shear force (*p* = 0.049). This corresponds with a higher (*p* < 0.001) cross-sectional area and muscle fibre diameter in these chickens. The highest content of n3 fatty acids (eicosapentaenoic, clupanodonic and docosahexaenoic acids) in breast meat were found in Dominant Sussex D104 chickens (*p* < 0.001). In contrast, a significantly higher α-tocopherol content (*p* < 0.001) and higher oxidative stability (*p* = 0.012) were found in Dominant Brown D102 (4.52 mg/kg and 0.282 mg/kg) and Dominant Tinted D 723 chickens (4.64 mg/kg and 0.273 mg/kg) in comparison with the Dominant Sussex D104 genotype (3.44 mg/kg and 0.313 mg/kg). The values of the atherogenic and thrombogenic indexes were the lowest (*p* < 0.001) in meat from Dominant Brown D102 chickens. Moreover, a lower cholesterol content (*p* < 0.001) was recorded from the genotypes Dominant Brown D102 (396 mg/kg) and Dominant Tinted D723 (306 mg/kg) chickens, contrary to the Dominant Sussex D104 cockerels (441 mg/kg). It can be concluded that cockerels Dominant Brown D102 are a suitable genotype for free-range rearing due to higher performance and higher pasture herbage intake, which positively influences meat quality, whereas the meat of Dominant Sussex D104 cockerels shows higher amounts of n3 fatty acids and lower n6/n3 ratios.

## 1. Introduction

In terms of welfare, fattening in alternative systems, such as free-range or organic systems, is preferable. According to Commission Regulation (EC) No 889/2008 [1], chickens in organic farming shall either be reared until they reach a minimum age (81 days) or else shall come from slow-growing strains. For this reason, fast-growing hybrids commonly used in intensive fattening are unsuitable for free-range and organic systems. Moreover, fast-growing chickens tend to be inactive and exhibit very low motor activity and foraging behaviour and thus do not benefit from large space allowances [2,3].

Medium- or slow-growing genotypes are recommended for alternative systems. The cockerels of the layer line or dual-purpose breeds are another suitable option. Moreover, caponisation is beneficial in these breeds, because the caponisation can increase the culinary quality of the meat [4,5] and bone strength [6] compared to cockerels. One-day-old male layers are usually culled directly after hatching due to their inefficient growth rate compared with broilers, which implies a long fattening period and unfavourable feed conversion [7]. The use of such cockerels in commercial production would be unprofitable. However, the culling of one-day-old egg-type cockerels is increasingly an ethical problem, and finding alternative solutions is desirable.

The functional meat properties of fast-growing and medium-growing strains appear much more suitable for both industry and the consumer (lower drip and cook losses and higher tenderness), whereas from a nutritional point of view, slow-growing meat appears healthier (less fat and a higher content of n3 polyunsaturated fatty acids (PUFA)) [3]. Dal Bosco et al. [8] found that the slow-growing strains (egg-type lines) seem to have a higher efficiency of eicosapentaenoic acid (EPA) and docosahexaenoic acid (DHA) deposition in comparison with meat-type chickens. This can be explained by the fact that elongation is partly affected by the oestrogen level [9]. In addition, as is evident from the study of Dal Bosco et al. [2], slow-growing chickens were more active and cover an average daily distance of 1,230 m, whereas fast-growing birds cover only 125 m. So, the slow-growing chickens possess a good aptitude for pasture. Access to pasture may provide fresh grass, insects and worms [10]. Lorenz et al. [11] estimated that approximately 10%–15% and 20%–25% of total feed intake may come from pasture in broilers and laying hens, respectively. This is equivalent to a dry matter pasture intake of 2 to 5 g per day. Pasture consumption and the typology of grass available can influence meat quality in free-range chickens by increasing vitamin, carotenoid and mineral content [12,13] and can also increase the levels of n3 PUFA in meat [8,14,15], with potential human health benefits but potentially lower oxidative stability [16]. However, there could also be an opposite effect because slower growth rates could lower oxidative stress by decreasing the metabolic production of free radicals [17].

The objective of this study was to compare performance characteristics and breast meat quality in the cockerels of three dual-purpose lines housed in mobile boxes on pasture. Meat quality was evaluated by determining shear force; muscle fibre characteristics and sensory analysis; vitamin and carotenoid content; and oxidative stability and fatty acid composition. The cockerels Dominant Sussex D 104, Dominant Brown D 102 and Dominant Tinted D 723 were used. These cockerels came from the Czech firm Dominant CZ, which specializes in breeding coloured layer hybrids for egg production and cockerels for meat. The strategy of the selection programme of this firm is to support the adaptability, flexibility and robustness in the bird for a variety of production conditions. Therefore, the Dominant genotypes are particularly suitable and popular in free-range and organic farms and are used in more than 45 countries on four continents.

## 2. Materials and Methods

### 2.1. Cockerels, Husbandry and Diets

The experiment was conducted with 300 one-day-old dual-purpose Dominant cockerels (Dominant CZ, Lázně Bohdaneč, Czech Republic). The chickens were divided into three groups (100 chickens per group) according to genotype: Dominant Sussex D 104, Dominant Brown D 102 and Dominant Tinted D 723. Chickens of all groups were kept in indoor pens until 49 days of age (stocking density: 0.067 m^2^/bird) on wood shavings with a 16 h lighting programme, gas heating and ventilation with a temperature-controlled fan. The temperature in the room at arrival was 32 °C and cotinuously decreased to 19.8 °C as the birds reached 49 days of age. Each pen was equipped with nipple drinkers and pan feeders. Chickens were relocated to floorless portable pens (0.108 m^2^/bird) on pasture until the end of the experiment at 77 days of age. The floorless portable pen dimensions were 3.0 × 3.6 × 0.6 m for a total area of 10.8 m^2^ per pen. The pen contained hat drinkers connected to a water basin. Feed was provided in trough feeders (length of 100 cm) with access on both sides. All three portable pens were constantly moved twice daily to restrict grassland damage. Once during the morning feeding at 8:00 h and again at 18:00 h. The dominant species of the pasture herbage were *Lolium perenne*, *Festuca pratensis*, and *Trifolium pratense*. Throughout the experiment, the chickens were fed three mixed feeds: a starter feed until 28 days of age, a grower feed between 29 and 70 days of age, and a finisher feed from 71 to 77 days of age. Feed and water were provided ad libitum. The ingredients and nutrient contents of the mixed feed and pasture herbage are shown in Table 1 and Table 2. The experiment lasted from May to July. Procedures performed with the animals were in accordance with the Ethics Committee of the Central Commission for Animal Welfare at the Ministry of Agriculture of the Czech Republic (Prague, Czech Republic) and carried out in accordance with Directive 2010/63/EU for animal experiments. The protocol of this experiment was approved by the Ethical Committee of the Institute of Animal Science (Prague-Uhříněves, Czech Republic), case number 04/2016.

The chickens were weighed at 0, 28, 49, 70 and 77 days of age. The number of chickens and their health status were checked twice a day during the experiment based on chickens activity, normal behaviour patterns (e.g., active feed and water intake, normal walking, wing stretching, calm and effortless breathing, energetic movements when distracted), voice, skin, plumage quality, stance and foot and limb formation. Feed intake was monitored daily on a per-pen basis. Feed conversion was calculated as the total feed intake divided by the total weight gain over the reporting period. Pasture intake was measured from the 50th to 77th day and indirectly assessed by the modified method of Dal Bosco et al. [18]. Pasture herbage samples were collected in square areas (50 × 50 cm) and then calculated for the whole area of each portable pen. The sward was first measured before the placement of the pen and once again after the portable pen was moved to another position.

At the end of the experiment, at 77 days of age, 16 chickens with an average body weight (average ± 50 g) were selected from each group and slaughtered. The chickens were slaughtered, and the breast muscles (*pectoralis major*) were dissected for analyses of meat quality (n = 8) and sensory analyses (n = 8).

### 2.2. Physical Analysis of the Breast Muscles

The ultimate pH values were detected 24 h postmortem using a 330i pH meter (WTW, Weilheim, Germany) with a glass probe introduced 1 cm deep into the transverse section of the *pectoralis major* muscle. Meat tenderness was determined by the Warner-Bratzler shear test in the fresh and boiled breast muscle. Samples of the pectoralis major muscle for determination of tenderness were packaged in plastic bags with zip ties and heated in a water bath at 75 °C for 1 h. The meat samples were cut into 2 × 1 cm cuboids with the cuts running parallel to the muscle fibres. Tenderness was measured using an Instron Model 3342 (Instron, Norwood, MA, USA) with a Warner-Bratzler shear blade with a triangular hole. The load cell was 20 N with a crosshead speed of 100 mm/min and a sampling rate of 20 points/s. The maximum shear force (N) was determined. The cooking loss was calculated from the differences between the weights of the raw and cooked samples.

### 2.3. Histochemical Analysis of Breast Muscles

To determine the histochemical parameters of the pectoralis major muscle, samples were collected immediately after slaughtering. The samples were frozen in 2-methylbutane cooled by liquid nitrogen (−156 °C) and stored at −80 °C until histochemical analysis. The samples were cut (cross-sections with a thickness of 12 μm) at −20 °C using a Leica CM 1850 cryostat (Leica Microsystems Nussloch GmbH, Nussloch, Germany). Subsequently, staining with haematoxylin and eosin for the basic histological characteristics of the muscle fibres was performed. Image analysis NIS Elements AR 3.1 (Laboratory Imaging s.r.o., Prague, Czech Republic) was used to detect the number of muscle fibres per 1 mm^2^, diameter, and fibre cross-sectional area.

### 2.4. Sensory Analysis of Breast Muscles

For sensory evaluation, the procedure described by Bureš et al. [19] was applied. Chicken breasts without skin were evaluated by a panel of ten selecteded assessors trained according to ISO 8586-1 [20]. The evaluation was performed in a sensory laboratory equipped with booths. The samples were cooked for 1 h at 180 °C without any spices or other ingredients. The samples were cut into approximately cubes (2 × 2 × 2 cm), placed in covered glass containers marked with three-digit random numbers and served at 50 °C to the sensory panel. Water and bread were provided to the panel members to neutralize their sensory precepts. The odour, tenderness, juiciness, flavor, and total acceptability of the samples were scored. A nine-point scale was used for the assessment (1–very undesirable, 9–very desirable).

### 2.5. Chemical Analyses of Breast Muscles

Samples of breast meat were stored until analyses in plastic bags at −20 °C. Dry matter was determined by drying in an oven (Memmert ULM 500; Memmert, Schwabach, Germany) at 105 °C to a constant weight. The ether extract was obtained by extraction with petroleum ether in a Soxtec 1043 apparatus (FOSS Tecator AB, Höganäs, Sweden). For determination of cholesterol in the meat, lipids were saponified, and the unsaponified matter was extracted with diethyl ether in accordance with ISO 3596:2011. Silyl derivatives were prepared using TMCS and HMDS silylation reagents (Sigma-Aldrich, Prague, Czech Republic) and quantified on a gas chromatograph equipped with a SAC-5 capillary column (Supelco, Bellefonte, PA, USA) that was operated isothermally at 285 °C. Fatty acid composition of the breast meat was determined after chloroform-methanol extraction of the total lipids [21]. Nonadecanoic acid (C 19:0) was used as an internal marker to quantify the FAs in the samples. Alkaline trans-methylation of the FAs was performed [22]. Gas chromatography of the methyl esters (FAMEs) was performed using an HP 6890 chromatograph (Agilent Technologies, Inc.) with a programmed 60 m DB-23 capillary column (150–230 °C) and a flame-ionisation detector; the split injections were performed using an Agilent autosampler. The fatty acids were identified by their retention times compared with standards. PUFA 1, PUFA 2, PUFA 3 and 37 Component FAME Mixes (Supelco, Bellefonte, PA, USA) were used as standards. The atherogenic index (AI) and the thrombogenic index (TI) were calculated in accordance with the methodology of Ulbricht [23]; and the peroxidation index (PI) was calculated according to Cortinas et al. [24]. The ratio between hypocholesterolemic and hypercholesterolemic fatty acids (hypocholesterolemic/hypercholesterolemic index; h/H) was calculated according to a formula mentioned in Santos-Silva et al. [25].

The contents of lutein and zeaxanthin were measured by high-performance liquid chromatography (HPLC) according to a modified method mentioned in the article of Froescheis et al. [26]. The HPLC instrument (VP series; Shimadzu, Kyoto, Japan) was equipped with a diode array detector. A Kinetex C18 column (100 × 4.6 mm; 2.6 µm; Phenomenex, Torrance, CA, USA) was used. A gradient system was applied with acetonitrile:water:ethyl acetate (88:10:2) as eluent A and acetonitrile:water:ethyl acetate (88:0:15) as eluent B.

The α-tocopherol and retinol contents in the skinless breast muscle were determined in accordance with the European standards EN 12,822 [27] and EN 12823-1 [28], respectively, by a Shimadzu HPLC system (VP series; Shimadzu, Kyoto, Japan) equipped with diode array detector. The samples were subjected to alkaline saponification with 60% potassium hydroxide followed by the appropriate extraction with diethyl ether.

The lipid peroxidation levels in the fresh breast meat and meat stored for 5 days at 4 °C were analysed by chromatographic analysis (a Shimadzu HPLC system (VP series; Shimadzu, Kyoto, Japan) equipped with a diode-array detector) using the modified method mentioned in the study of Czauderna et al. [29]. The column utilized was a Phenomenex C18 column (Synergi 2.5 µm, Hydro-RP, 100 Å, 100 × 3 mm) (Phenomenex, Torrance, USA). Solvent A consisted of water-acetonitrile (95:5, *v/v*), and solvent B consisted of acetonitrile. The lipid oxidative stability was expressed as mg of malondialdehyde (MDA) per kg of meat.

### 2.6. Chemical Analyses of Diet and Freeze-Dried Pasture

The dry matter of the diet and freeze-dried pasture was determined by drying in an oven at 105 °C to a constant weight. The ether extract was obtained by extraction with petroleum ether in a Soxtec 1043 apparatus (FOSS Tecator AB, Höganäs, Sweden) and the protein content was determined using a Kjeltec Auto 1030 Analyser (Tecator, Höganäs, Sweden). The contents of fatty acids, vitamins, and carotenoids were analysed using the methods described previously.

### 2.7. Statistical Analyses

The data were analysed using one-way analysis of variance (ANOVA) with the general linear models (GLM) procedure in SAS software [30]. The main effect was the genotype of the chickens. All differences were considered to be significant at *p* < 0.05. The results in the tables are presented as the mean and standard error of the mean (SEM).

## 3. Results

The performance characteristics of the Dominant chickens are summarized in Table 3. Significant differences in body weight were recorded for the duration of the experiment. At the end of the experiment (77th day), the Dominant Brown D 102 chickens had the highest (*p* < 0.001) body weight (1842 g). The lowest body weight (*p* < 0.001) was observed in the Dominant Tinted D 723 chickens (1613 g). Additionally, values related to feed intake, feed conversion and herbage intake were not statistically evaluated because there were no replications. Feed intake was the lowest in these chickens. However, the Dominant Brown D 102 chickens showed the lowest value of feed conversion. Higher pasture herbage intake was observed in Dominant Brown D 102 and Dominant Tinted D 723 chickens. 

The Dominant Tinted D 723 cockerels had a significantly (*p* = 0.001) higher pH measured 24 h *postmortem* compared to the other two genotypes (Table 4). Additionally, the tenderness of the boiled meat was higher in Dominant Tinted D 723 chickens. This is evident from both the values of shear force measured using the Warner-Bratzler test (*p* = 0.049) and from the sensory evaluation (*p* = 0.022). Tenderness of the meat is also related to muscle fibres. Muscle fibre type IIB (white, fast glycolytic fibres) was evaluated, because only this fibre type is found in the breast muscle of chickens. The genotype did not affect the number of muscle fibres per 1 mm^2^. However, the cross-sectional area (*p* < 0.001) and diameter (*p* < 0.001) of the muscle fibres were significantly influenced. The highest values were observed in the Dominant Brown D102 (1410 µm^2^ and 40.6 µm) and Dominant Tinted D 723 (1430 µm^2^ and 40.9 µm) chickens, contrary to the Dominant Sussex D104 (1312 µm^2^ and 39.0 µm). The highest value of tenderness, evaluated on the basis of sensory analysis, was recorded in the breast meat of Dominant Tinted D 723 chickens (6.16), and the lowest value was from the Dominant Brown D 102 (5.35).

As shown in Table 5, carotenoid storage in breast meat was not influenced by genotype. However, significantly (P<0.001) higher α-tocopherol content was ascertained in Dominant Brown D 102 (4.52 mg/kg) and Dominant Tinted D 723 chickens (4.64 mg/kg), contrary to genotype Dominant Sussex D 104 (3.44 mg/kg). Additionally, higher oxidative stability in fresh breast meat (*p* = 0.012) was found in these first two groups (0.282 and 0.273 vs. 0.313 mg of malondialdehyde per kg of meat).

Genotype had an effect on most fatty acids in breast meat (Table 6). The meat of Dominant Sussex D 104 chickens showed higher (*p* < 0.001) contents of myristic, palmitic, margaric, stearic and arachidic acids as saturated fatty acids. In the case of monounsaturated fatty acids, a higher (*p* < 0.001) content of oleic and erucic fatty acids was found in Dominant Sussex D 104 chickens, whereas Dominant Brown D 102 chickens had a higher (*p* < 0.001) content of palmitoleic fatty acids in breast meat. For polyunsaturated fatty acids, linoleic (*p* = 0.018), eicosadienic (*p* < 0.001), arachidonic (*p* < 0.001), eicosapentaenoic (*p* < 0.001), clupanodonic (*p* < 0.001) and docosahexaenoic (*p* < 0.001) fatty acid contents were higher in the breast meat of Dominant Sussex D 104 chickens, and the α-linolenic fatty acid content (*p* = 0.013) was higher in Dominant Brown D 102 chickens. The n6/n3 ratio was under 5 in all groups. The lowest value (*p* < 0.001) was found in Dominant Sussex D 104 chickens (4.31). In terms of meat quality in relation to human health, the mutual ratios of fatty acids, which are expressed as indexes, are important. The atherogenic and thrombogenic indexes were reduced (*p* < 0.001) in meat from Dominant Brown D 102 chickens. The hypocholesterolemic/hypercholesterolemic fatty acid ratio, which considers the specific effects of fatty acids on cholesterol metabolism, was the highest (*p* = 0.008) in Dominant Brown D 102 chickens, and a higher value is more desirable. This finding corresponds to the cholesterol content, which was demonstrably lower (*p* < 0.001) in Dominant Brown D 102 (396 mg/kg) and Dominant Tinted D 723 (306 mg/kg) chickens, contrary to the Dominant Sussex D 104 genotype (441 mg/kg).

## 4. Discussion

The growth ability of dual-purpose chickens is very small in comparison with broilers [31]. In the present study, the highest body weight on the 77th day of the experiment (1842 g) and the nonsignificantly lowest feed conversion (2.79) was found in Dominant Brown D 102 cockerels. Data concerning the performance of cockerels Dominant are limited. Therefore, the results will be compared especially with the values of egg-type or dual-purpose chickens. In the case of indoor fattening, Dominant Sussex D 104 cockerels weighed 1974 g on the 77th day [32]. This is 178 g more than in the same genotype in the present experiment. In the case of dual-purpose lines, male Tetra-H chicks from indoor and free-range housing systems weighed 1790 g after 70 days of fattening [33] and Bresse Gauloise raised under a welfare-enhanced organic system that includes access to pasture weighed 2570 g after 84 days [34]. The feed conversion of Dominant Sussex D 104 cockerels (0–84 days) in the study of Jelínková [32] was 3.1. These values are comparable to the results of the present study.

The greatest ultimate pH was found in the meat of the Dominant Tinted D 723 cockerels. Glamoclija et al. [35] and Mueller et al. [31] showed that meat pH was lower in broilers fattened for a longer time or in slow-growing genotypes. Berri et al. [36] attained similar findings. The increasing growth rate and/or muscle development could slow down the postmortem glycolysis and increase the ultimate pH by lowering the glycogen content of the breast muscle [37]. The pH value relates to cooking loss and tenderness because a low pH is associated with a deteriorated water-holding capacity of fresh meat. In the present study, the tenderness evaluated using the Warner-Bratzler blade and sensory analysis was the highest in the Dominant Tinted D 723 cockerels. However, no significant differences were observed in cooking loss. Negative effects of the outdoor rearing system were found in a study by Michalczuk et al. [38]. Sirri et al. [3] reported use of egg-type slow-growing birds negatively influenced the water-holding capacity and texture of the meat. Debut et al. [39] and Fanatico et al. [40] also found higher drip and cook losses in the breast meat of slow-growing birds reared outdoors than in fast-growing broilers. Tenderness generally decreases as chickens become older [41].

In relation to the tenderness of meat, it is important to monitor muscle fibre characteristics. In accordance with the present study, Lukasiewicz et al. [42] consistently reported that muscle fibre size is influenced by genotype. The total number of fibres was 15%–20% higher in the cockerels of the fast-growing line than in those of the slow-growing line [43]. Chicken breast muscles are entirely composed of white fibres. A higher white muscle fibre thickness has a positive effect on meat tenderness and a negative effect on juiciness [44]. However, in the experiment carried out by Yang et al. [45], the opposite result was obtained. The meat of cockerels with the highest muscle fibre diameter showed the highest water-holding capacity and the highest shear force value.

The significant contribution of forage in increasing antioxidants in eggs and meat is evident from many studies [12,46,47,48,49]. In the present study, the pasture vegetation (analysed in the form of freeze-dried pasture) was especially rich in the carotenoids lutein and zeaxanthin (175.0 and 170.3 mg/kg DM, respectively), and the detected amount was higher than in the abovementioned works. Forage intake is affected by the motor activity of chickens and is influenced by outdoor enrichment and season [48]. Therefore, the higher content of α-tocopherol in the breast meat of Dominant Brown D 102 and Dominant Tinted D 723 chickens may be related to the higher consumption of pasture vegetation in these chickens. Genotype Dominant Brown D 102 reached 7.41 g dry matter (DM)/bird/day on the 70th day of fattening and Dominant Tinted D 723 cockerels reached 7.52 g DM/bird/day, in contrast to Dominant Brown D 102 chickens (5.95 g DM/bird/day). Skřivan et al. [50] and Dal Bosco et al. [48] ascertained an almost two-fold higher α-tocopherol content in the meat of chickens with outdoor access. Vitamin E and carotenoids present in pasture vegetation usually reduce the sensitivity of unsaturated fatty acids to oxidation. The higher presence of vitamin E, as an antioxidant, in meat also increased its oxidative stability in the present experiment. Contrary to the results of the present study, Dal Bosco et al. [48] and Cartoni Mancinelli et al. [49] showed that despite the improved antioxidant profile of the free-range group, the oxidative processes of the meat increased. This effect was probably due to the higher kinetic activity of the outdoor chickens and the resulting process of oxidation [51]. On the other hand, the results of the study of Mattiolli et al. [52] suggested that moderate exercise in birds with innate locomotory behaviour is beneficial because it produces a lower dose of radicals that enhance non-enzymatic antioxidant defences (vitamin E and retinol). The oxidative stability of meat is also influenced by its lipid profile. Polyunsaturated fatty acids, whose source is pasture vegetation, are more easily oxidized than saturated fatty acids [48].

In the present study, rapeseed oil, which is rich in unsaturated fatty acids, was also used as a source of fat in the mixed feed; therefore, the breast meat of all three groups met the World Health Organization requirement for an n6/n3 ratio of 5. The feed mixture contained palmitic, stearic, oleic and linoleic fatty acids, whereas the pasture herbage was rich in α-linolenic fatty acid content. This fact was reflected in the fatty acid composition in the breast meat. The genotype Sussex D 104, which showed lower pasture herbage consumption, had higher contents of palmitic, stearic, oleic and linoleic fatty acids and lower contents of α-linolenic fatty acids in breast meat compared to the two other genotypes. Improvement in the fatty acid profile of meat due to the higher availability of polyunsaturated n3 fatty acids through pasture intake is also evident from the studies of Ponte et al. [14], Dal Bosco et al. [48], and Michalczuk et al. [15]. On the other hand, Givens et al. [53] and Funaro et al. [54] stated that a free-range system with pasture vegetation access decreased the n3 fatty acid content and increased the n6/n3 ratio in breast meat. Moreover, slow-growing genotypes show a high percentage of polyunsaturated fatty acids and a decrease in the n6/n3 ratio compared to fast-growing hybrids [8]. Almost two-fold higher contents of arachidonic, eicosapentaenoic, clupanodonic, and docosahexaenoic fatty acids were observed in Dominant Sussex D 104 chickens compared to the other two groups. Although the amount of these fatty acids (except clupanodonic acid) was comparable in both mixed feed and pasture vegetation. This result can be explained by the findings of Alessandri et al. [9], who stated that in the case of slow-growing strains of egg-type lines, a higher content of EPA and DHA is expected because laying hens seem to have a higher deposition efficiency of these fatty acids with respect to meat-type chickens, being that elongation is partly affected by the estrogen level. In addition, Lizardo et al. [55] assumed that 15% of dietary fatty acids are converted through metabolic processes. Birds are able to desaturate and elongate α-linolenic acid to eicosapentaenoic acid and docosahexaenoic fatty acid [56,57]. Slow-growing chicken breasts contained approximately 2- to 3-fold higher amounts of long-chain n3 fatty acids and arachidonic acid, respectively, than those in medium- and fast-growing strains [8], which suggests a different expression of genes encoding for the desaturating enzymes [3,58]. The higher content of both n6 and n3 polyunsaturated fatty acids in the breast meat of slow-growing chickens reflected the increased expression/activity of the Δ5 and Δ6 desaturase [58]. In addition, slow-growing birds showed higher kinetic activities that produced different metabolism [8]. Therefore, the levels of n3 and n6 fatty acids in meat were probably influenced by the genetic foundation of chickens of egg type lines. Dominant Sussex D 104 is the result of crossing of two Sussex lines, whereas Dominant Brown 102 is the result of crossing the Rhode Island Red paternal line with the Rhode Island White maternal line and Dominant Tinted D 723 is a product of crossing the White Leghorn paternal line with the Rhode Island Red maternal line.

In terms of the product’s impact on human health, the fatty acid indexes are more meaningful than the individual fatty acid levels alone. For example, the atherogenic and thrombogenic indexes reflect the probability of pathogenic phenomena such as atheroma and thrombus formation. Meat consumption of Dominant Brown D 102 chickens reduces the risk of occurrence of these phenomena the most. A lower cholesterol content in meat was also detected in this genotype as well as in Dominant Tinted D 723 chickens. This reduction could be due to a higher forage intake. As previously shown by Ponte et al. [59], the inclusion of leguminous forages in broiler diets contributed to the decreased cholesterol content of broiler meat. Pasture herbage is a good source of tocotrienols [60] and tocotrienols help lower plasma cholesterol levels [61]. In another study, Ponte et al. [14] stated that consumption of the leguminous pasture had a marginal effect on the cholesterol content of broiler meat.

## 5. Conclusions

Slow-growing chickens, which show higher physical activity and thus the possibility of grazing, are used in free-range systems. Dual-purpose breeds represent a possible alternative for this fattening method. The meat of dual-purpose cockerels fattened on pasture herbage is more mature and enriched in antioxidants and n3 fatty acids, contrary to conventionally fattened fast-growing chickens, and thus, it can be considered as a functional food.

From the three tested Dominant genotypes, Dominant Brown D 102 cockerels are the most suitable genotype for free-range fattening due to nonsignificantly higher performance and higher pasture herbage intake, which positively influenced meat quality. The cockerels Dominant Brown D 102 achieved on the 77th day of age a weight of 1842 g and a feed conversion 2.79. The breast meat of this genotype was characterized by higher α-tocopherol content and oxidative stability and lower atherogenic and thrombogenic indexes and cholesterol content compared to two other genotypes. On the other hand, the meat of Dominant Sussex D104 cockerels showed more n3 fatty acids and a lower n6/n3 ratio. It can be concluded that, the cockerels of these Dominant genotypes are able to fully exploit the potential of pasture herbage for improving the quality of their meat.

## Figures and Tables

**Table 1 animals-10-00387-t001:** Ingredients of the diets.

Ingredient (g/kg)	Starter	Grower	Finisher
Soybean meal	360.0	248.0	215.0
Maize	277.5	210.0	210.0
Wheat	290.0	420.0	486.7
Wheat bran	-	50.0	39.6
Rapeseed oil	30.0	30.0	18.0
Sodium chloride	3.0	3.0	3.0
Monocalcium phosphate	13.0	11.0	7.5
Limestone	17.0	18.5	12.5
L-Lysine hydrochloride	1.3	2.1	1.0
DL-Methionine	2.9	2.1	1.7
L-Threonine	0.3	0.3	-
Vitamin-mineral premix ^1^	5.0	5.0	5.0

^1^ vitamin-mineral premix provided per kg of diet: retinyl acetate 3.6 mg, cholecalciferol 13 μg, α-tocopherol acetate 30 mg, menadione 3 mg, thiamine 3 mg, riboflavin 5 mg, pyridoxine 4 mg, cyanocobalamin 40 μg, niacin 25 mg, calcium pantothenate 12 mg, biotin 0.15 mg, folic acid 1.5 mg, choline chloride 250 mg, copper 12 mg, iron 50 mg, iodine 1 mg, manganese 80 mg, zinc 60 mg, selenium 0.3 mg.

**Table 2 animals-10-00387-t002:** Nutrient content of the diets and freeze-dried pasture.

Analysed Nutrient Content (g/kg)	Starter	Grower	Finisher	Freeze Dried Pasture Herbage
Dry matter (g/kg)	887	890	885	924
Crude protein (g/kg)	203	171	161	116
Fat (g/kg)	23.59	24.50	19.60	32.46
AME (by calculation MJ/kg)	12.5	12.1	11.9	3.8
SFA (mg/100 g)	843	1144	853	616
MUFA (mg/100 g)	1608	1693	1566	262
PUFA (mg/100 g)	1774	1733	1736	1609
n6/n3	5.41	5.04	6.59	0.260
α-Tocopherol (mg/kg)	44.9	34.8	33.0	52.6
Retinol (mg/kg)	2.06	2.95	1.98	-
Zeaxanthin (mg/kg)	0.640	0.600	0.592	157.4
Lutein (mg/kg)	0.976	0.910	0.809	161.7

AME = apparent metabolizable energy, SFA = saturated fatty acids (caproic, caprylic, capric, lauric, tridecylic, myristic, pentadecylic, palmitic, margaric, stearic, arachidic, heneicosylic, behenic, tricosylic), MUFA = monounsaturated fatty acids (myristoleic, palmitoleic, oleic, vaccenic, 11-eicosenoic, erucic, nervonic), PUFA = polyunsaturated fatty acids (linoleic, γ-linolenic, α- linolenic, eicosadienoic, dihomo- γ-linolenic, arachidonic, eicosatrienoic, eicosapentaenoic, docosapentaenoic, docosahexaenoic).

**Table 3 animals-10-00387-t003:** Performance traits of Dominant cockerels.

Characteristic	Dominant			SEM	Probability
	Sussex D 104	Brown D 102	Tinted D 723		
**Body Weight (g)**					
Day 0	40.1 ^a^	40.0 ^a^	38.7 ^b^	0.20	0.008
Day 28	430 ^b^	442 ^a^	402 ^c^	2.5	<0.001
Day 49	951 ^a^	965 ^a^	874 ^b^	4.6	<0.001
Day 70	1578 ^b^	1629 ^a^	1415 ^c^	9.0	<0.001
Day 77	1796 ^b^	1842 ^a^	1614 ^c^	10.2	<0.001
**From 0 to 49th Day**					
Feed intake (g/day/bird)	43.0	38.3	41.2		
F:G (g/g)	2.32	2.04	2.37		
**From 50th to 77th Day**					
Feed intake (g/day/bird)	93.8	98.5	87.5		
F:G (g/g)	3.81	3.95	3.99		
**From 0 to 77th Day**					
Feed intake (g/day/bird)	59.8	58.1	56.8		
F:G (g/g)	2.91	2.79	3.01		
**Average Pasture Herbage Intake (g DM/Day/Bird)**					
From 50th to 63rd day	4.65	6.40	5.69		
From 64th to 77th day	5.95	7.41	7.52		

F:G = feed:gain, DM = dry matter, SEM = standard error of the mean. ^a,b,c^ means with different superscripts differ significantly.

**Table 4 animals-10-00387-t004:** Physical characteristics, muscle fibre characteristics and sensory analysis of breast meat.

Characteristic	Dominant			SEM	Probability
	Sussex D 104	Brown D 102	Tinted D 723		
pH_24_	5.62 ^b^	5.66 ^b^	5.78 ^a^	0.019	0.001
Cooking loss (%)	22.1	20.2	21.5	0.51	NS
**Warner-Bratzler shear force (N)**					
Fresh meat	18.14	16.75	17.74	0.558	NS
Boiled meat	39.44 ^a^	37.92 ^a,b^	33.46 ^b^	1.036	0.049
**Muscle fibre (type IIB) characteristics**					
Number of fibres (per 1 mm^2^)	557	512	521	17.9	NS
Cross-sectional area (μm^2^)	1312 ^b^	1410 ^a^	1430 ^a^	12.8	<0.001
Diameter (μm)	39.0 ^b^	40.6 ^a^	40.9 ^a^	0.22	<0.001
**Sensory analysis ^1^**					
Odour	5.74	5.64	5.61	0.110	NS
Tenderness	5.69 ^a,b^	5.35 ^b^	6.16 ^a^	0.121	0.022
Juiciness	5.53	5.40	5.85	0.104	NS
Flavour	5.80	5.61	5.94	0.109	NS
Total acceptability	5.89	5.50	6.05	0.110	NS

^1^ all traits were assessed by a 10-member panel on a scale from 1 (very undesirable) to 9 (very desirable). SEM = standard error of the mean, NS = not significant. ^a,b^ means with different superscripts differ significantly.

**Table 5 animals-10-00387-t005:** Carotenoid and vitamin contents and oxidative stability of fat in breast meat.

Characteristic	Dominant			SEM	Probability
	Sussex D 104	Brown D 102	Tinted D 723		
Lutein (mg/kg)	0.158	0.139	0.203	0.0123	NS
Zeaxanthin (mg/kg)	0.139	0.130	0.193	0.0120	NS
Retinol (mg/kg)	0.040	0.041	0.044	0.0016	NS
α-Tocopherol (mg/kg)	3.44 ^b^	4.52 ^a^	4.64 ^a^	0.148	<0.001
MDA (day 0, mg/kg)	0.313 ^a^	0.282 ^b^	0.273 ^b^	0.0061	0.012
MDA (day 5, mg/kg)	0.366	0.372	0.356	0.0093	NS

SEM = standard error of the mean, MDA = malondialdehyde, NS = not significant. ^a,b^ means with different superscripts differ significantly.

**Table 6 animals-10-00387-t006:** Composition and indexes of fatty acids, fat content and cholesterol content in breast meat.

Characteristic		Dominant			SEM	Probability
		Sussex D 104	Brown D 102	Tinted D 723		
**Fatty Acid (mg/100 g)**						
Myristic	C 14:0	11.37 ^a^	4.20 ^b^	3.18 ^c^	0.780	<0.001
Palmitic	C 16:0	196 ^a^	143 ^b^	121 ^b^	8.3	<0.001
Margaric	C 17:0	2.27 ^a^	1.26 ^b^	1.17 ^b^	0.122	<0.001
Stearic	C 18:0	102.8 ^a^	54.0 ^b^	55.0 ^b^	5.50	<0.001
Arachidic	C 20:0	1.22 ^a^	0.79 ^b^	0.65 ^b^	0.065	<0.001
Palmitoleic	C 16:1	15.8 ^b^	20.8 ^a^	10.0 ^c^	1.16	<0.001
Oleic	C 18:1	268 ^a^	207 ^b^	161 ^c^	12.0	<0.001
Eicosenoic	C 20:1	1.78	2.11	1.54	0.132	NS
Erucic	C 22:1	0.083 ^a^	0.055 ^b^	0.044 ^b^	0.0045	<0.001
Linoleic	C 18:2	91.2 ^a^	81.4 ^a,b^	66.6 ^b^	3.74	0.018
α-Linolenic	C 18:3	2.77 ^b^	3.10 ^a^	2.95 ^b^	0.094	0.013
γ-Linolenic	C 18:3	7.19	8.67	6.49	0.324	NS
Eicosadienic	C 20:2	2.69 ^a^	2.00 ^b^	1.65 ^c^	0.105	<0.001
Arachidonic	C 20:4	47.5 ^a^	28.5 ^b^	23.4 ^c^	2.28	<0.001
Eicosapentaenoic	C 20:5	2.42 ^a^	1.55 ^b^	1.23 ^b^	0.135	<0.001
Clupanodonic	C 22:5	11.06 ^a^	6.73 ^b^	6.30 ^b^	0.548	<0.001
Docosahexaenoic	C 22:6	13.10 ^a^	6.70 ^b^	5.27 ^b^	0.766	<0.001
**Sums and Ratios of Fatty Acids**						
SFA		319 ^a^	206 ^b^	184 ^b^	14.6	<0.001
MUFA		304 ^a^	246 ^b^	185 ^c^	13.6	<0.001
PUFA		183 ^a^	143 ^b^	117 ^c^	7.4	<0.001
n3		34.4 ^a^	24.1 ^b^	19.7 ^b^	1.56	<0.001
n6		148 ^a^	118 ^b^	97 ^c^	5.8	<0.001
n6/n3		4.31 ^b^	4.91 ^a^	4.95 ^a^	0.077	<0.001
h/H		2.13 ^b^	2.32 ^a^	2.15 ^b^	0.029	0.008
AI		0.500 ^a^	0.412 ^c^	0.449 ^b^	0.0087	<0.001
TI		0.941 ^a^	0.789 ^b^	0.910 ^a^	0.0171	<0.001
**Fat (g/kg DM)**		8.50	9.50	9.69	0.386	NS
**Cholesterol (mg/kg)**		441 ^a^	396 ^b^	306 ^c^	13.0	<0.001

DM = dry matter, SEM = standard error of the mean, NS = not significant. ^a,b^ means with different superscripts differ significantly. SFA = saturated fatty acids, MUFA = monounsaturated fatty acids, PUFA = polyunsaturated fatty acids, AI = atherogenic index, TI = thrombogenic index, h/H = hypocholesterolemic/hypercholesterolemic fatty acid ratio.

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
