# Peer review of "Performance and Meat Quality of Dual-Purpose Cockerels of Dominant Genotype Reared on Pasture"

_animals, 2020, doi:10.3390/ani10030387_

Round 1

Reviewer 1 Report

Authors evaluated the performance, meat characteristics and traits of three dual-purpose chicken genotypes on pasture.

Some suggestions:

-Title: The word "dominant" is confusing, specify what it is.

-Simple summary and Abstract: Both starts with layer/egg-type chicken but the study was conducted with dual-purpose genotype. Please modify it.

- Table 2.: There were no replicates for feed intake, feed conversion, herbage intake as authors used only one pen per genotype. Consequently without statistical analyses, one does not know there is a difference or not between groups. Please modify sentences according to this.

Author Response

Answer: Dear reviewer, we would like to thank you for the comments of the article and requirements for its correction. All of your requirements were beneficial for the final quality of this article. We have accepted all of your requirements and adjusted the text according to them. Adjustments of the text were made in accordance with the requirements of other reviewers. All changes in the manuscript are marked in yellow.

Authors evaluated the performance, meat characteristics and traits of three dual-purpose chicken genotypes on pasture.

Some suggestions:

-Title: The word "dominant" is confusing, specify what it is.

Answer: The title of the manuscript was rewritten. The term Dominant was specified. It is a name of genotype.

-Simple summary and Abstract: Both starts with layer/egg-type chicken but the study was conducted with dual-purpose genotype. Please modify it.

Answer: The “Simple summary” and “Abstract” were enriched.

- Table 2.: There were no replicates for feed intake, feed conversion, herbage intake as authors used only one pen per genotype. Consequently without statistical analyses, one does not know there is a difference or not between groups. Please modify sentences according to this.

Answer: Values related to feed intake, feed conversion and herbage intake were not statistically evaluated because there were no replications. This information was added into the text and sentences were modified.

Reviewer 2 Report

See attached PDF file

Author Response

Answer: Dear reviewer, we would like to thank you for the comments of the article and requirements for its correction. All of your requirements were beneficial for the final quality of this article. We have accepted your requirements and adjusted the text according to them. Adjustments of the text were made in accordance with the requirements of other reviewers. All changes in the manuscript are marked in yellow.

Manuscript title: The Meat Quality of the Dual-purpose Cockerels Dominant Reared on Pasture

General Comments: The authors have submitted a manuscript that outlines interesting research on alternative uses for byproduct cockerel chicks. I do believe this work is worthy of publication in Animals, but I would recommend a significant amount of changes and/or clarifications before it is published. My comments and suggestions are outlined below:

Title: The syntax for the title could be improved. I’d suggest “Performance and meat quality of dualpurpose Dominant cockerels reared on pasture” as an alternative.

Answer: The title of the manuscript was rewritten as you suggested.

Lines11, 12, 17, and other places throughout manuscript: I’d suggest using an alternative term to “fattening.” For most readers this term relates to livestock species and not necessarily poultry. I’d suggest using “finish” and/or “finishing.”

Answer: The term “fattening” is a commonly used term in poultry.

Line 16: “...and increase the quality of their meat” seems like a statement that the birds are making a conscious decision; I’d suggest “...with resulting improvements in meat quality.”

Answer: The sentence was rewritten as you suggested.

Line 19: “an” should go in front of “ethical”

Answer: “an” was added.

Line 20: I’d suggest changing “find” to “assess”

Answer: The words were changed.

Line 22: It is somewhat unclear if “n=100" refers to total number of birds or number of birds per strain used in this study.

Answer: The information was added.

Line 24: Typical industry verbiage would be to report density as area/bird (e.g., m2/bird)

Answer: The density was expressed in m2/bird.

Line 27: I’d suggest an alternate term for “ascertained” in this sentence; what does “pc” in “dry matter/pc/day”?

Answer: The term “ascertained” was replaced by recorded”, “pc” was replaced by “bird”.

Line 49-50: These two sentences need improvement. Stating “there is an increasing interest from people regarding animal welfare” seems overly declarative.

Answer: The sentences were rewritten.

Line 51: Can a reference be provided for this?

Answer: The reference was added.

Line 61: I’d suggest avoiding adjectives such as “great,” “good,” “excellent,” etc.

Answer: The word was replaced.

Lines 63-66: Suggest editing this sentence; what does “attractive” mean in this context?

Answer: The term was replaced.

Line 67: Change “for” to “of”

Answer: It was changed.

Line 69: I am unsure of the context for “...elongation is partly affected by the oestrogen level” in this sentence.

Answer: The sentence was rewritten.

Line 69-71: I would suggest clarifying that this information is from a previous study. As stated, it seems overly declarative. Locomotion of chickens would be dramatically influenced by the setting.

Answer: The sentence was rewritten.

Line 71-72: I’d suggest changing to “Access to pasture may provide...”

Answer: The sentence was rewritten as you suggested.

Line 74: Is it 2 to 5 g per day?

Answer: Yes, it is. This information was added.

Line 75: Remove “the”

Answer: “The” was removed.

Line 76: Comma after “[4, 10, 11]”

Answer: The was comma was added.

Line 77: Remove “...of the meat”

Answer: of the meat” was removed.

Line 82: Remove “the”; add ‘;’ after “force” and after “analysis” instead of commas

Answer: The sentence was rewritten as you suggested.

Line 83: Add ‘;’ after “content”

Answer: The sentence was rewritten as you suggested.

Line 86: What does “festive meals” mean?

Answer: The term “festive meals” was replaced by term “meat”.

Line 96: Suggest m2/bird instead (other locations in the manuscript also)

Answer: The density was expressed in m2/bird.

Line 97: Brooding and room/house temperatures should be added

Answer: The temperature in room during indoor housing of chickens was added.

Line 98: Remove “Then the...”

Answer: “Then the...” was removed.

Line 101: Suggest removing “chicken”

Answer: “chicken” was removed.

Line 102: How pen movement occurred daily? Was it the same for all pen? Was it consistent daily? What parameters dictated the daily pen movement?

Answer: The information about movement of pens was added.

Table 1: I’d suggest separating this into two tables...one for ingredient composition of diets and one for nutrient composition of diets and herbage. It seems odd having a table with blank space for herbage where the ingredient composition is shown.

Answer: Table 1 was separated to two tables.

Line 121: How specifically was health status monitored?

Answer: The information was added into the text.

Line 120-125: Was mortality (%) monitored? How was feed conversion determined? Was feed conversion adjusted for mortality?

Answer: The mortality was monitored. The information about feed conversion determination was added into the text. Feed conversion was adjusted for mortality (dead pieces were weighed and added to total body weight).

Line 126: Not sure what “...of average weight” specifically means. Were birds selected based on a certain deviation from sample mean?

Answer: The sentence was rewritten.

Line 133-134: Remove “the” before “determination” and before “tenderness”; remove “of boiled meat”

Line 149: Remove “the” before “breast meat”

Line 150: Remove first “the” in sentence

Line 161-162: Remove both “the” beginning these sentences; remove “of the meat”

Line 165: Remove “the” before “lipids”

Line 169: Remove “the” before “fatty acid”

Answer: The words were removed.

Lines 239-242: Were the observed differences statistically significant?

Answer: Values related to feed intake, feed conversion and herbage intake were not statistically evaluated because there were no replications. This information was added into the text. And sentences related to these characteristics were modified.

Table 2: I would suggest changing title to “Performance traits of Dominant cockerels”; why were no SEMs and P-values given for feed intake and feed conversion?

Answer: The title was changed. Values related to feed intake, feed conversion and herbage intake were not statistically evaluated because there were no replications. One group = one pen.

Line 250-251: Authors state “...muscle fiber type IIB...was found in the breast muscle” ... does this statement refer to findings in this study or something different?

Answer: The sentence was reformulated.

Line 298: Remove “the” before “data”

Line 299: Remove “very”

Answer: The words were removed.

Lines 302-310: Authors state these values are comparable to their results. It seems the performance data indicated here from previous research is not completely comparable to the present study. Birds were grown for different lengths of time and likely under different management environments. May want to focus on those studies that are the closest in length and management conditions as the current study.

Answer: The text was modified according to requirement.

Line 319: Remove “the” before “cooking loss” and before “negative”

Line 320: Remove “that the” before “use”

Answer: The words were removed.

Line 325: Avoid “good”

Answer: The word was changed.

Line 333: Where are these antioxidant effects seen?

Answer: The sentence was enriched.

Line 346: Avoid using first person words such as “our”

Answer: The word was replaced.

Line 371: Change “were” to “was”

Answer: The word was changed.

Lines 368-376: This sentence is extremely long; suggest editing

Answer: The sentence was modified.

Line 398: Remove “are known to”

Answer: “are known to” was removed.

Line 402-404: First sentence is confusing; suggest rewriting to make more succinct

Answer: The sentence was rewritten.

Lines 406-407: I’m not sure what “...it can be interesting for many consumers” means; needs more specificity

Answer: The sentence was modified.

Line 410: Change “influence” to “influenced”

Consider adding a more definitive recommendation and/or concluding statement for your readers.

Answer: The definitive concluding statement was added.

Reviewer 3 Report

Dear Authors,

all comments included in attachment. 

It is interesting manuscript about the alternative use of unwanted cockreals from laying production. 

You should fix some elements in the manuscript.

But generally, there are minor corretions.

Kind Regards,

Reviewer

Author Response

Answer: Dear reviewer, we would like to thank you for the comments of the article and requirements for its correction. All of your requirements were beneficial for the final quality of this article. We have accepted your requirements and adjusted the text according to them. Adjustments of the text were made in accordance with the requirements of other reviewers. All changes in the manuscript are marked in yellow.

  1. The word was changed.

  2. The word was changed.

  3. The information about caponisation was added into the text.

  4. There were no replications. One group = one pen. Therefore, values related to feed intake, feed conversion and herbage intake were not statistically evaluated.

  5. The table was modified.

  6. Individual fatty acids belonging to SFA, MUFA and PUFA were listed below the table.

  7. The word was changed.

  8. The word was changed.

  9. There were no replications. One group = one pen. Therefore, values related to feed intake, feed conversion and herbage intake were not statistically evaluated.

  10. We would like to apologize, but the collagen content in breast muscles was not analyzed.

Round 2

Reviewer 2 Report

Authors appear to have made substantial changes to improve the readability of the manuscript. I do have concerns over study design with lack of replications.